# Data-Driven Insights into Labor Progression with Gaussian Processes

**Tilekbek Zhoroev** [1,2,†] 🆔, **Emily F. Hamilton** [1,3,†] 🆔 **and Philip A. Warrick** [1,4,*,†] 🆔

1 Medical Research and Development, PeriGen Inc., Cary, NC 27518, USA
2 Department of Applied Mathematics, North Carolina State University, Raleigh, NC 27606, USA
3 Department of Obstetrics and Gynecology, McGill University, Montreal, QC H3A 0G4, Canada
4 Department of Biomedical Engineering, McGill University, Montreal, QC H3A 0G4, Canada
* Correspondence: philip.warrick@perigen.com
† These authors contributed equally to this work.

**Abstract:** Clinicians routinely perform pelvic examinations to assess the progress of labor. Clinical guidelines to interpret these examinations, using time-based models of cervical dilation, are not always followed and have not contributed to reducing cesarean-section rates. We present a novel Gaussian process model of labor progress, suitable for real-time use, that predicts cervical dilation and fetal station based on clinically relevant predictors available from the pelvic exam and cardiotocography. We show that the model is more accurate than a statistical approach using a mixed-effects model. In addition, it provides confidence estimates on the prediction, calibrated to the specific delivery. Finally, we show that predicting both dilation and station with a single Gaussian process model is more accurate than two separate models with single predictions.

**Keywords:** labor; obstetrics; cardiotocography; electronic fetal monitoring; biomedical signals; signal processing; gaussian processes

## 1. Introduction

The progression of labor is a critical aspect of childbirth and is a central focus in obstetric research and clinical practice. Labor progression disorders (arrest of cervical dilation or fetal descent) are the leading indications for intrapartum cesarean delivery [1]. Models of normal cervical dilation over time have served as references for expected progress. Clinical guidelines that define arrest of cervical dilation are based on deviations from these curves or their related statistics regarding the time to advance one centimeter in dilation [2,3].

The natural advance of labor toward delivery involves a sequence of physiological changes and developments that are assessed in standard clinical practice by periodic pelvic examinations. These assessments involve documentation of various parameters including cervical dilation, cervical effacement, fetal station, and the examination time. The fact that the timing of these exams is not fixed, coupled with the diversity in lengths of labor, presents significant challenges to developing a labor model based on these observations.

Traditional approaches to monitoring labor progression have revolved around the passage of time only, which, although beneficial, have limitations in adaptability and precision. The Friedman curve [4] was the first of these and it had a profound influence on clinical decision making. The more recent polynomial model of Zhang et al. [5] was more data-driven and provided improved estimates of cervical dilation. However, the compliance of clinicians with the guidelines based on this model is low [6] and has not been associated with a reduction in cesarean-section rates. This suggests that clinicians consider other factors that affect dilation and descent, beyond time alone, which can lead to overriding the guidelines. Moreover, these methods often fail to capture the non-linear, multifaceted nature of maternal and fetal indicators that influence the progression of labor. In contemporary biomedical research, there is an increasing inclination towards employing

machine learning models that can learn from the data, accommodating inter- and intra-subject variability—such as those that exist across and within deliveries–to provide a personalized healthcare approach.

Within this context, Gaussian Process (GP) models emerge as a promising tool. Originating from the Bayesian non-parametric family of models, GP models provide a flexible and robust approach to modeling relationships between input and output variables, accommodating uncertainty and noise within individual data. The literature is rapidly expanding with applications of GPs in various domains, reflecting its flexibility and robustness in handling various kinds of data [7–9]. In particular in healthcare care, GP models have been noted for their ability to provide probabilistic predictions and naturally adapt to uncertainty and noise within individual data [10–14]. GPs have been effectively used for probabilistic predictions in Alzheimer's disease [13], modelling intrapartum uterine pressure and fetal heart rate [14], and critical care monitoring, where they have been shown to predict clinical interventions [11]. Furthermore, GP models have also shown promise in the field of epidemiology, particularly in the predictive modeling of infectious diseases [15]. This wide array of applications in healthcare underscores the adaptability of GP models to various medical data challenges. In a clinical context, model predictions and their uncertainty estimates are crucial for the decision-making process because they provide clinicians with an informed rationale for using or disregarding the model prediction. However, one significant limitation that hinders the widespread application of standard GP models is their computational complexity, which scales cubically with the size of the dataset, thereby posing challenges in scenarios where real-time decision-making is crucial.

Sparse Gaussian Process Regression (SGPR) models, an evolution of standard GP models, are engineered to tackle such computational challenges [16–19]. By introducing inducing points or a set of virtual observations, these models enable a significant reduction in computational cost, which is particularly beneficial for large-scale problems. The sparse methods approximate the full GP model, maintaining a balance between computational efficiency and model performance.

In this paper, we develop a new model to predict dilation and station using a Gaussian processes approach, and use SGPR to apply it to a large database of labor progress. Because GPs model the probability distributions of the input and output variables, they have the desirable attribute of including an estimate of the uncertainty associated with each prediction, based on the observed delivery data. Inspired by [20] and documented determinants of cervical dilation, we formulate this as a regression problem to predict, as outputs, the dilation and station of the current exam, using as inputs the observations of the dilation, effacement, and station of the previous exam. We also include as an input predictor the cumulative contraction count derived from automated analysis of the uterine pressure (UP) signal. Finally, we include other clinical information about induction and administration of epidural as input predictors. Importantly, unlike the Friedman and Zhang curves, the model is causal, using only past information for prediction, and is therefore more suitable for real-time clinical use during labor.

## 2. Methods

### 2.1. Data

Data came from 49,694 births between 2017 and 2021 in 10 US hospitals in Ohio. We first selected deliveries with a gestational age of at least 35 weeks and having a live singleton fetus in vertex presentation. Further inclusion criteria included nulliparity, vaginal delivery, and an Apgar score at 5 min of 7 or more. In addition, deliveries were required to have more than one cervical exam in the first stage of labor and to have recorded UP from cardiotocography, also known as electronic fetal monitoring (EFM). We excluded records with shoulder dystocia or with admission to the maternal or newborn ICU using data from the electronic medical record (EMR, Epic™, Verona, WI, USA). Applying these criteria resulted in 8022 births with 47,714 associated pelvic exams, which we used for constructing the models.

A second independent dataset from a different U.S. region was used for validation. These data came from consecutive deliveries in 2021 from 6 hospitals in Oregon and Washington. Of the 5528 total deliveries in this set, 527 births with 2378 associated pelvic exams satisfied the criteria for inclusion in the validation set.

The data for each pelvic exam included the cervical dilation, cervical effacement, fetal station, and examination time. For each delivery, we had access to the EFM record (PeriWatch Cues™, PeriGen, Cary, NC, USA). We also used other EMR data including the use of induction, the time of epidural administration, and the time of rupture of membranes. The full clinical characteristics of the data are described in a more clinically focused description of this work in [21].

From a myriad of potential variables that affect cervical dilation, our experimentation was based on key variables with documented evidence of their relationship to cervical dilation. Professional guidelines list three essential elements to consider when assessing the adequacy of cervical dilation in the first stage of labor: the duration of unchanged dilation, the nature of uterine contractions, and the degree of cervical dilation attained [1–3]. Cervical effacement and fetal descent also affect cervical dilation [22–24]. We also studied the effect of three interventions that can affect labor progression: use of labor induction, presence of epidural anesthesia, or rupture of the amniotic membrane. At this time, we did not examine the role of maternal race/ethnicity or maternal factors such as BMI. The amount of data needed for modeling grows steeply with the number of predictors under study in order to find sufficient examples of all factor combinations.

### 2.2. Preprocessing

We restricted the analysis to the last 20 h of labor before delivery. After matching corresponding UP and EMR records, we deidentified the data while transforming all times to be relative to the time of the first pelvic exam in the 20-h window, which we denote $t_0$. We removed exams that had descending dilation, or occurred within 5 min after the previous exam. We linearly interpolated any missing observation of dilation, effacement, or station for exams that had both previous and successive exams with valid corresponding observations. Otherwise, these exams were also removed. Contractions were detected using automated analysis of the EFM records to determine their time extent (PeriWatch Cues™, PeriGen, Cary, NC, USA). From this detection, we calculated the cumulative number of contractions since $t_0$ at the time of each pelvic exam. Finally, based on the timing of membrane rupture and epidural administration, each pelvic exam was given categorical attributes for the presence of each of these two conditions.

### 2.3. Gaussian Processes

A Gaussian process is the appropriate extension of a multivariate Gaussian distribution to a Gaussian distribution over a particular family of functions. In the same way that a sample from an $n$-dimensional Gaussian distribution is an $n$-dimensional vector, a sample from a Gaussian process is a random function. Gaussian distributions are specified by their mean vector and covariance matrix; similarly, a Gaussian process is completely specified by a mean function and a covariance function (sometimes referred to as a kernel), which can be evaluated at any point, according to the following formal definition.

**Definition 1.** *A Gaussian process (GP) is a collection of random variables, any finite number of which have a joint Gaussian distribution.*

Mathematically, a Gaussian process is defined as follows,

$$f \sim \mathcal{GP}(\mu_\phi(\mathbf{x}), k_\theta(\mathbf{x}, \mathbf{x}'))$$
$$\mu_\phi(\mathbf{x}) = \mathbb{E}[f(\mathbf{x})]$$
$$k_\theta(\mathbf{x}, \mathbf{x}') = \mathbb{E}[(f(\mathbf{x}) - \mu_\phi(\mathbf{x}))(f(\mathbf{x}') - \mu_\phi(\mathbf{x}'))]$$

where $\phi$ and $\theta$ represent the parameters of the mean and covariance functions, respectively, for the Gaussian process $f$. Assume that we have given input vectors $\mathbf{X} = \{\mathbf{x}_1, \mathbf{x}_2, \ldots, \mathbf{x}_n\}$ and observations $\mathbf{y} = \{y_1, y_2, \ldots, y_n\}$ with Gaussian noise variance $\sigma^2$, and that $y(\mathbf{x})|f(\mathbf{x}) \sim \mathcal{N}(y(\mathbf{x}); f(\mathbf{x}), \sigma^2)$. Then, by the definition of the Gaussian processes, the function values $f(\mathbf{X})$ have a joint Gaussian distribution,

$$f(\mathbf{X}) = [f(\mathbf{x}_1), f(\mathbf{x}_2), \ldots, f(\mathbf{x}_n)]^\top \sim \mathcal{N}(\boldsymbol{\mu}_\phi(\mathbf{X}), K_\theta^{\mathbf{XX}})$$

with a mean vector, $(\boldsymbol{\mu}_\phi(\mathbf{X}))_i = \mu_\phi(\mathbf{x}_i)$, and covariance matrix $(K_\theta^{\mathbf{XX}})_{ij} = k_\theta(\mathbf{x}_i, \mathbf{x}_j)$. As a result, the posterior predictive distribution Gaussian process on test data $\mathbf{X}_*$ is given by

$$f(\mathbf{X}_* | \mathbf{X}_*, \mathbf{X}, \mathbf{y}, \sigma^2) \sim \mathcal{N}(\boldsymbol{\mu}^*, \boldsymbol{\Sigma}^*)$$
$$\boldsymbol{\mu}^* = \boldsymbol{\mu}_\phi(\mathbf{X}_*) + K_\theta^{\mathbf{X}_*\mathbf{X}}(K_\theta^{\mathbf{XX}} + \sigma^2\mathbf{I})^{-1}(\mathbf{y} - \boldsymbol{\mu}_\phi(\mathbf{X}))$$
$$\boldsymbol{\Sigma}^* = \sigma^2\mathbf{I} + K_\theta^{\mathbf{X}_*\mathbf{X}_*} - K_\theta^{\mathbf{X}_*\mathbf{X}}(K_\theta^{\mathbf{XX}} + \sigma^2\mathbf{I})^{-1}K_\theta^{\mathbf{XX}_*}.$$

It is worth noting that $(K_\theta^{\mathbf{X}_*\mathbf{X}})^\top = K_\theta^{\mathbf{XX}_*}$ and it represents the covariance matrix evaluated for each $\mathbf{x}_i$, $\mathbf{x}_{*j}$ pair.

In numerous modeling scenarios, the selection of an appropriate mean and covariance (kernel) function is pivotal. The mean function, often chosen as a linear function of inputs, represents the expected value of the process and serves as a baseline around which variations are modeled. It is crucial to provide a sensible starting point for the model, especially when prior knowledge about the expected behavior of the underlying process is available. A well-chosen mean function can significantly enhance the predictive accuracy of the model and reduce the amount of data required for reliable predictions.

Additionally, much focus is placed on the selection of an appropriate covariance function, denoted $k_\theta(\cdot, \cdot)$. This function is responsible for transforming pairs of input values, $\mathbf{x}_i, \mathbf{x}_j$, into a constant, $k_\theta(\mathbf{x}_i, \mathbf{x}_j)$, which delineates the covariances between pairs of random variables, $f(\mathbf{x}_i), f(\mathbf{x}_j)$. The covariance function is paramount because it encodes our prior beliefs about the function we are modeling, reflecting assumptions about the smoothness, periodicity, and other properties of the function. It acts as a statistical surrogate model of the function, determining the degree of influence one observation can have on another, thereby shaping the overall behavior of the Gaussian Process model.

To qualify as a covariance function, any chosen function must produce a positive, semi-definite covariance matrix, ensuring the resulting Gaussian Processes are well-defined. Several commonly employed covariance functions are Gaussian (also known as squared exponential or radial basis function), rational quadratic, periodic, polynomial, and the Matérn class of covariance functions. The choice of the specific form of the kernel function is not arbitrary; it is a critical decision that conveys our prior beliefs and assumptions about the statistical characteristics of the function being modeled.

### 2.4. GP Model Description and Kernel Selection

The objective of this research was to estimate cervical dilation and fetal station, two key indicators of labor progression, based on delivery data that are routinely recorded. One of the challenges of modelling with our large dataset of delivery records was the computational burden. We chose a Sparse Gaussian Process Regression (SGPR) [19] model because it is well suited to large datasets with complex relationships between variables. It uses a sparse approximation of the covariance matrix, which reduces computational complexity without unduly sacrificing accuracy.

Despite the large number of deliveries in the overall dataset, the data available per delivery are relatively sparse. This scarcity of individual delivery data led us to conceptualize each delivery as a distinct meta-task, according to the methodologies and insights of the article [25]. This method is crucial as it allows the model to integrate prior knowledge from each unique delivery meta-task, thereby improving its adaptability and predictive accuracy

across a wide range of delivery data. This, in turn, enables a more nuanced characterization of the progression of individual delivery.

We chose to use a neural network to express the mean function of our model. This approach, inspired and validated by [25], empowers the model to unravel the complex inherent patterns and relationships within the data. Consequently, this facilitates more nuanced and precise predictions. Our selection of kernels was a strategic amalgamation of Gaussian, Matérn, and linear kernels, each chosen for its unique attributes and relevance to our dataset. The Gaussian kernel helps smooth out the function, which is important because our data points on cervical dilation and fetal station are continuous. This kernel is particularly relevant to our dataset as it allows for the modeling of smooth transitions and progressions, which are plausibly related to clinical observations of dilation and station. On the other hand, the Matérn kernel provides the much-needed flexibility to model non-differentiable functions and it is adept at accommodating the inherent variability and irregularities within the labor progression data, reflecting the diverse and unpredictable nature of individual labor progressions. Furthermore, the inclusion of the linear kernel promotes long-term non-decreasing behavior, which is critical in modeling labor progression. In the long run, cervical dilation and fetal station follow a non-decreasing trajectory, and the linear kernel contributes to the ability of the model to maintain this physiological phenomenon.

To achieve our objective of concurrently predicting dilation and station, it was essential to multiply the selected kernels by a coregionalization kernel [26]. This approach facilitated the effective modeling of the correlation between the two critical outcomes, dilation, and station, ensuring coherent and mutually informed predictions, which is crucial for an understanding of labor progression.

*2.5. Model Implementation*

We implemented our Gaussian Process model using the GPFlow library [27], a Python-based framework for GP modeling. The integration of GPFlow with TensorFlow enabled efficient computations and was important when using advanced optimization algorithms. We defined our SGPR model with a deep mean function and a composite kernel consisting of Gaussian, Matérn, and linear kernels.

In our study, the set of meta-tasks $M$ consists of datasets corresponding to each delivery, $M = \{D_i\}_{i=1}^{m}$, with each dataset $D_i$ containing observations $\{x_i, y_i\}$. In this setting, all meta-tasks share the same input and output dimensionalities, but they can have different numbers of observations. The meta-learning framework then aims to learn a GP prior from these meta-tasks, which can be adapted quickly to data from a new delivery not seen during training.

The mean function of our SGPR was parameterized using deep neural networks, as described in the previous section. The training of this model involves optimizing the SGPR hyperparameters, including those of the mean function, to maximize the marginal likelihood of the meta-tasks.

During testing, for a new delivery (or meta-task), the learned SGPR prior was fine-tuned using the data for that delivery. This involved updating the SGPR hyperparameters using the test delivery data, resulting in a posterior distribution that was used for the predictions. This approach ensured that the model is informed by the broader patterns learned across multiple deliveries and the specific nuances of the individual delivery data.

We used a hybrid optimization strategy to optimize the model parameters, combining the strengths of two optimization algorithms: (1) Adam and (2) limited memory Broyden Fletcher Goldfarb Shanno with simple bounds (L-BFGS-B). Adam is an extension of stochastic gradient descent that is effective in handling non-convex optimization landscapes. L-BFGS-B is a quasi-Newton method that is well-suited to optimizing smooth functions in high-dimensional spaces. This hybrid approach facilitated efficient convergence to the optimal solutions in the high-dimensional parameter space of our SGPR model. After

optimization, we evaluated the performance metrics of the model on the test data from the cross-validation.

### 2.6. Sparse Gaussian Process Regression Model Input Variables

In the strategic construction of our Sparse Gaussian Process Regression (SGPR) model, a selection of key variables has been integrated to effectively predict labor progression. The input variables for our SGPR model were chosen from the selection process described in Section 3.2. They included previous dilation, previous station, previous effacement, cumulative contraction counts, presence of epidural anesthesia, and an indicator that denotes whether labor was induced or occurred spontaneously.

One of the primary variables is the previous cervical dilation, derived from historical data, which serves as a fundamental indicator of how labor has progressed over time. Complementing this, the model also considers the previous fetal station, which provides insight into the baby's position in the birth canal, a crucial factor in determining the progress of labor. Another critical element is the previous cervical effacement. The previous effacement adds a significant layer of detail to the model, enhancing its ability to understand the dynamics and rate of progression of labor [22–24].

The accuracy of the model is significantly enhanced by the inclusion of three key variables, determined through multiple simulations and statistical comparisons. First, cumulative contraction counts are incorporated, acknowledging the pivotal role of uterine contractions in childbirth. This variable, which includes the frequency of contractions and effectively quantifies a primary driver of labor. Moreover, the model incorporates information on whether labor was induced or occurred spontaneously and presence of epidural anesthesia. The inclusion of these three variables has been statistically shown to improve the accuracy and robustness of the model, providing a more comprehensive and reliable tool for understanding labor progression.

### 2.7. Mixed-Effects Model

Mixed-effects (ME) models are a statistical approach to analyzing hierarchical data, which is data that are nested or structured at multiple levels. In the obstetrics research area, mixed-effects models can be used to account for the variability in birth outcomes that is due to both individual and group-level factors. We chose this statistical model to provide a comparison to the GP models we developed in this study.

Accordingly, we used mixed-effects models based on [20] to analyze data from multiple births recorded across various hospitals over different periods. Our ME model, implemented using the statsmodels Python package [28], is designed to handle complex statistical computations and define both fixed and random parameters explicitly, addressing the nested nature of our data. The model equation is formulated as follows:

$$
\begin{aligned}
\text{Dilation} = {} & \beta_0 + \beta_1 \times (\text{previous dilation}) + \beta_2 \times (\text{previous station}) \\
& + \beta_3 \times (\text{previous effacement}) + \beta_4 \times (\text{cumulative contraction count}) \\
& + \beta_5 \times (\text{epidural}) + D_{(\text{Delivery ID})} + \epsilon
\end{aligned}
$$

In this equation, $\beta_0, \beta_1, \ldots, \beta_5$ represent the coefficients of the fixed effects: previous dilation, previous station, previous effacement, cumulative contraction count, and epidural usage. The term $D_{\text{Delivery ID}}$ denotes the random effect associated with each delivery. The error term $\epsilon$ is assumed to follow a normal distribution with mean zero and variance $\sigma^2$, i.e., $\epsilon \sim \mathcal{N}(0, \sigma^2)$.

The fixed effects included the variables chosen from the selection process described in Section 3.2. Conversely, the random effects in our model capture the unexplained variability in birth outcomes, potentially stemming from individual factors like the mother's health or the baby's position during birth. These random effects are crucial to understanding the unique characteristics of each birth and to capture broader patterns in the data.

The model summary, Table 1, indicated significant results with all fixed effects showing strong associations with the dilation outcome. The model converged successfully, suggesting a good fit to the data.

**Table 1.** Parameter estimation of the fixed effect coefficients and the random effect covariance for the mixed-effects (ME) model.

| Name | Coefficient | Standard Error | *p*-Value |
|---|---|---|---|
| Fixed Effects ($\beta$) | | | |
| Previous Dilation ($\beta_1$) | 0.734 | 0.004 | <0.001 |
| Previous Station ($\beta_2$) | 0.077 | 0.007 | <0.001 |
| Previous Effacement ($\beta_3$) | 0.026 | 0.000 | <0.001 |
| Cumulative Contraction Count ($\beta_4$) | 0.001 | 0.000 | <0.001 |
| Epidural ($\beta_5$) | 0.613 | 0.015 | <0.001 |
| Random Effects ($D$) | | | |
| Group Variance (Delivery ID) | 0.140 | 0.011 | – |
| Model variance ($\sigma^2$) | 2.035 | – | – |

### 2.8. Performance Evaluation

For each ME and GP model, we evaluated performance using a 10-fold cross-validation approach. In each iteration, the model was trained on 80% of the data, validated on 10% of the data, and tested on the remaining 10% of the data. We ensured that for each fold, every data point was used in exactly one of the training, validation, and testing partitions and that each data point was tested in exactly one of the folds, providing a comprehensive assessment of the model's capabilities across diverse data scenarios. We considered one model superior to another if it had a lower average root mean square error (RMSE) in predicting the test set of each fold.

## 3. Results

### 3.1. Observed Labor Progression Trajectories

In this section, we present observed trajectories of normal labor progression, focusing on two factors. Figure 1a,b showcase examples of dilation and station trajectories, respectively. To facilitate comparison in the presentation of these and subsequent figures, we transform the horizontal time axis to be relative to the time of delivery. A close examination of these figures reveals the various paths that labor can take, from consistent and smooth progression to instances of abrupt change in dilation or fetal station. The variability in these trajectories underscores the unpredictable nature of childbirth.

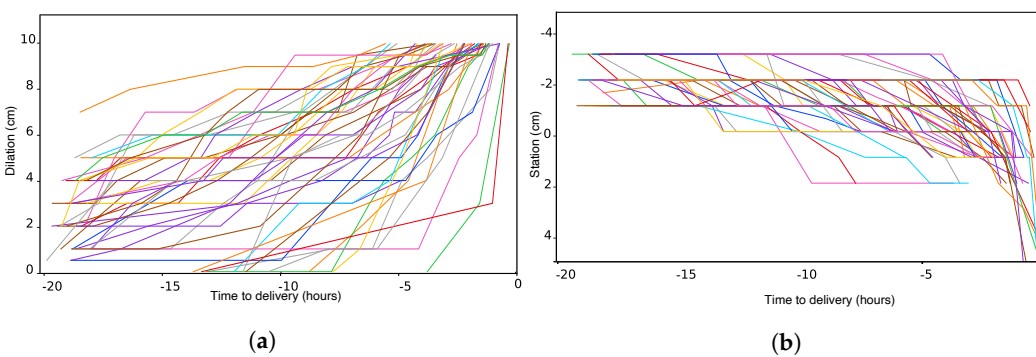

**(a)**          **(b)**

**Figure 1.** Variability of (**a**) dilation and (**b**) station trajectories over the course of labor. Each colored contour corresponds to an individual delivery.

### 3.2. Predictor Selection

Based on serial 10-fold cross-validation experiments, the SGPR model of dilation that had the lowest mean RMSE included the following variables: cumulative contraction count since $t_0$, dilation, effacement, and station in the previous exam, presence of epidural anesthesia, and use of induction. We used these predictors to obtain the results for the GP and ME multivariate models described below.

### 3.3. Sample Predicted Labor Progression

In Figure 2, we compare model performance graphically in two individual clinical examples—one with normal labor progression ending in vaginal delivery and the other with very abnormal labor progression ending with cesarean delivery for arrest of dilation. Figure 2a,c show the SGPR time-alone and multivariate models, respectively, applied to the normal labor example. Figure 2b,d show the same two models applied to the abnormal labor example. The SGPR model with multiple predictors of dilation stands out in its ability to adapt and precisely capture variations across a spectrum of labor trajectories. Whether dilation follows a steady course, exhibits fast transitions, or is interrupted with periods of arrest, the model consistently demonstrates its robustness and adaptability.

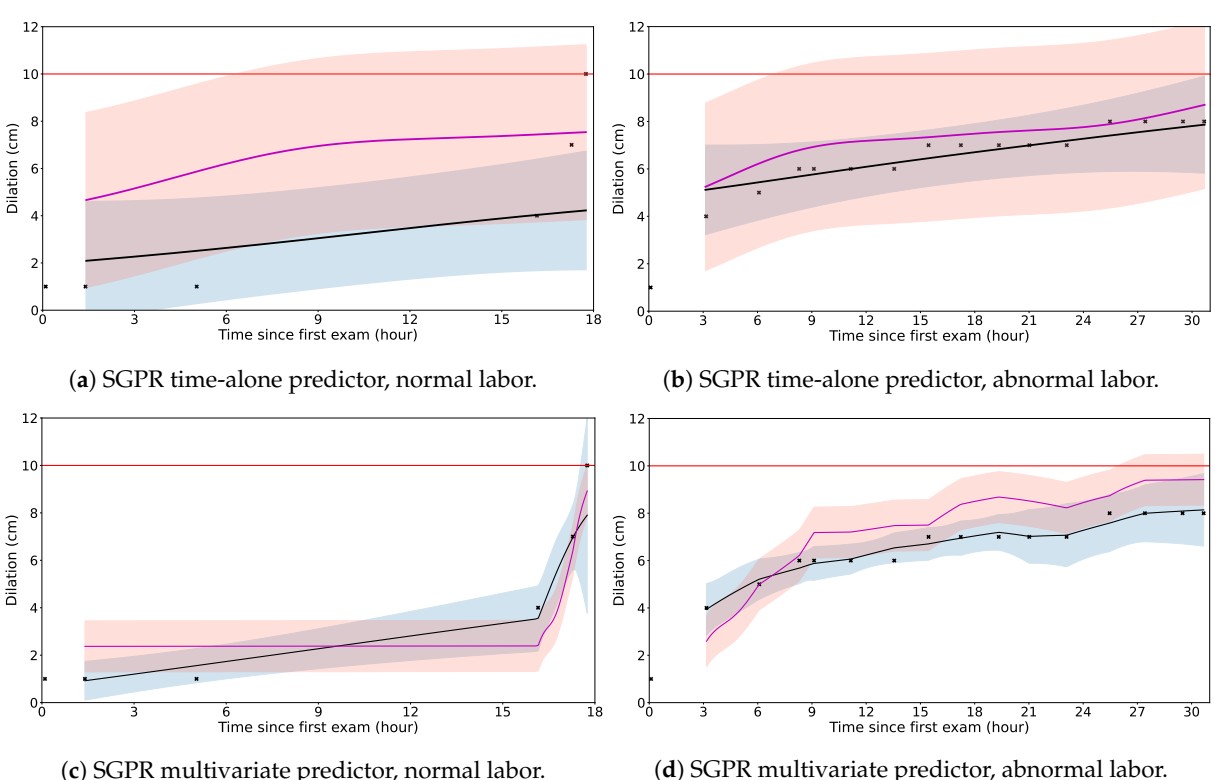

(**a**) SGPR time-alone predictor, normal labor.

(**b**) SGPR time-alone predictor, abnormal labor.

(**c**) SGPR multivariate predictor, normal labor.

(**d**) SGPR multivariate predictor, abnormal labor.

**Figure 2.** Examples of normal (left, **a**,**c**) and abnormal labor (right, **b**,**d**) from the dataset using a time-alone (top, **a**,**b**) and a multivariate SGPR predictor (bottom, **c**,**d**). The black crosses (x) are the observed dilation measurements. The pink line is the SGPR (standard) mean prediction; the black line is the SGPR (meta-trained with test set adaptation) mean prediction; the pink band is the SGPR (standard) 90% prediction interval; the blue band is the SGPR (meta-trained with test set adaptation) 90% prediction interval.

### 3.4. Population-Level Analysis of Labor Progression Prediction

This section transitions from individual case studies to a population analysis, synthesizing the predictive power of our SGPR model on a population scale. Central to our discussion is Figure 3, which depicts the mean predictions for dilation and station over time, enveloped by a 90% confidence interval, and the corresponding prediction errors, respectively. Figure 3a represents the expected progression of dilation, and Figure 3b repre-

sents the expected progression of fetal descent (station) at the population level, as predicted by the multivariate GP model. The graphical representation is comprehensive, with the plotted mean values providing a clear indication of the most probable trajectories of the progression of the labor. Complementing this, the 90% interval encapsulates the variability and uncertainty inherent in labor progression, acknowledging that while patterns can be discerned, the individual experiences can diverge significantly from the mean.

On the contrary, Figure 3c,d provide a critical evaluation tool that illustrates the prediction error over time. These highlight instances where the model's predictions align closely or deviate from the recorded observations.

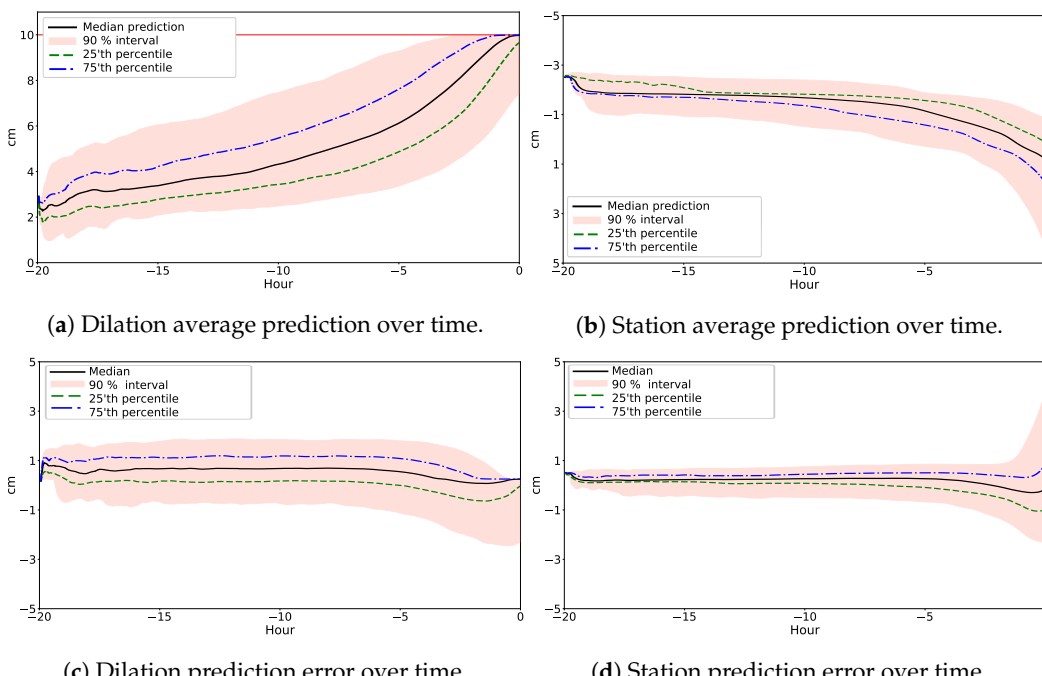

(**a**) Dilation average prediction over time.　　(**b**) Station average prediction over time.

(**c**) Dilation prediction error over time.　　(**d**) Station prediction error over time.

**Figure 3.** Population-based predictive analysis of the SGPR model using multiple predictors. These figures collectively illustrate the predictive power of our model at a population level, encapsulating both the expected progression and the corresponding prediction errors for dilation and station. The mean predictions, enveloped by a 90% confidence interval, underscore the most probable predicted labor trajectories, while the error plots critically assess the model's alignment with actual recorded data, highlighting its accuracy and reliability in diverse labor scenarios.

### 3.5. Evaluative Comparison of Predictive Models

This section represents the performance and comparison of various labor progression prediction models. Table 2 compares the RMSE values of different models, assessed over 10-fold cross-validation on the Ohio dataset. The models include $ME_{dil}$, a mixed-effects dilation predictor, and four GP models. The first GP model is a basic time-based model $GP(t)_{dil}$ predicting dilation. The others were multivariate including $GP_{dil}$ which forecasts dilation, $GP_{stat}$ which forecasts station, and $GP_{dil+stat}$ which predicts both dilation and station concurrently. For $ME_{dil}$ and the multivariate GPs, we use the predictors selected in Section 3.2.

Both multivariate models $ME_{dil}$ and $GP_{dil}$ outperformed $GP(t)_{dil}$, highlighting the value of incorporating various fixed and random effects. Model $GP_{dil+stat}$, which predicted both dilation and station simultaneously, outperformed all other dilation models. In particular, while station prediction for $GP_{dil+stat}$ compared to $GP_{stat}$ was not significantly different, the superior dilation prediction of $GP_{dil+stat}$ was statistically significant, highlighting the synergistic advantage of a simultaneous predictive approach.

Table 3 compares the RMSE values of the 10 cross-validation models on the second independent test set, in the same format as Table 2. The performance of the GP model was

consistent across both datasets, affirming its robustness and generalizability. In this case, $GP_{dil+stat}$ prediction was superior, with statistical significance, to both $GP_{dil}$ for dilation and $GP_{stat}$ for station. For $GP_{dil+stat}$, we show both Model $5_p$, the standard prediction of SGPR, and Model $5_b$, the prediction of SGPR with adaptation to the test set.

Because the GP model with test adaptation $5_b$ had some RMSE errors that were inferior to $5_p$, which had no adaptation, we were motivated to further understand their error characteristics. The absolute error distributions for a single cross-validation fold are shown for the Ohio dataset (Figure 4b) and the independent test set (Figure 4b). We found that these distributions were representative of those in all the other folds as well. Model $5_b$ had a stronger peak at lower errors, compared to model $5_p$. However, model $5_b$ also had some relatively infrequent, but large errors that approached the right limit of the horizontal axes ($\sim$5 cm and 12 cm, respectively).

Table 4 compares the mean absolute error (MAE) for models $5_p$ and $5_b$ for the Ohio and independent datasets. For both datasets, model $5_b$ had a lower prediction error, with statistical significance, compared to $5_p$ for both dilation and station prediction. This was a striking contrast to the same comparison with RMSE in Tables 2 and 3, indicating that the infrequent but large $5_b$ prediction errors weighted the RMSE, which masked the generally lower $5_b$ errors demonstrated by the distributions of Figure 4.

**Table 2.** Comparison of model prediction RMSE errors on cross-validation test sets for the Ohio data. The name subscript indicates the prediction output(s). For GP($t$), time was the sole model predictor.

| Model | Name | Dilation RMSE (cm) | vs. Model | *p*-Value | Station RMSE (cm) | vs. Model | *p*-Value |
|---|---|---|---|---|---|---|---|
| 1 | $GP(t)_{dil}$ | $2.504 \pm 2.382 \times 10^{-2}$ | - | - | - | - | - |
| 2 | $ME_{dil}$ | $1.176 \pm 1.252 \times 10^{-2}$ | 1 | $<1 \times 10^{-4}$ * | - | - | - |
| 3 | $GP_{dil}$ | $1.168 \pm 1.163 \times 10^{-2}$ | 2 | 0.141 | - | - | - |
| 4 | $GP_{stat}$ | - | - | - | $0.6625 \pm 1.535 \times 10^{-3}$ | - | - |
| $5_p$ | $GP_{dil+stat}$ | $1.126 \pm 1.057 \times 10^{-2}$ | 3 | $<1 \times 10^{-4}$ * | $0.6601 \pm 7.518 \times 10^{-3}$ | 4 | 0.7474 |
| $5_b$ | $GP_{dil+stat}$ | $1.093 \pm 3.51 \times 10^{-2}$ | $5_p$ | $1.3 \times 10^{-2}$ * | $0.7276 \pm 1.05 \times 10^{-2}$ | $5_p$ | $<1 \times 10^{-4}$ * |

* null hypothesis (model performances were not different) rejected at the $p < 0.05$ level.

**Table 3.** Comparison of model prediction root-mean squared (RMS) errors on the second independent data.

| Model | Name | Dilation RMSE (cm) | vs. Model | *p*-Value | Station RMSE (cm) | vs. Model | *p*-Value |
|---|---|---|---|---|---|---|---|
| 1 | $GP(t)_{dil}$ | $2.661 \pm 8.120 \times 10^{-4}$ | - | - | - | - | - |
| 2 | $ME_{dil}$ | $1.382 \pm 1.015 \times 10^{-3}$ | 1 | $<1 \times 10^{-4}$ * | - | - | - |
| 3 | $GP_{dil}$ | $1.424 \pm 1.699 \times 10^{-3}$ | 2 | $<1 \times 10^{-4}$ * | - | - | - |
| 4 | $GP_{stat}$ | - | - | - | $0.8687 \pm 4.70 \times 10^{-4}$ | - | - |
| $5_p$ | $GP_{dil+stat}$ | $1.354 \pm 1.469 \times 10^{-2}$ | 3 | $<1 \times 10^{-4}$ * | $0.8499 \pm 4.46 \times 10^{-4}$ | 4 | $<1 \times 10^{-4}$ * |
| $5_b$ | $GP_{dil+stat}$ | $1.400 \pm 1.310 \times 10^{-2}$ | $5_p$ | $<1 \times 10^{-4}$ * | $0.9004 \pm 4.50 \times 10^{-3}$ | $5_p$ | $<1 \times 10^{-4}$ * |

* null hypothesis (model performances were not different) rejected at the $p < 0.05$ level.

**Table 4.** Comparison of GP model prediction mean-absolute (MA) errors on the Ohio (O) and independent (I) datasets, without (Model $5_p$) and with (Model $5_b$) test adaptation.

| Data | Model | Name | Dilation MAE (cm) | vs. Model | *p*-Value | Station MAE (cm) | vs. Model | *p*-Value |
|---|---|---|---|---|---|---|---|---|
| O | $5_p$ | $GP_{dil+stat}$ | $0.826 \pm 8.68 \times 10^{-3}$ | - | - | $0.512 \pm 4.55 \times 10^{-3}$ | - | - |
| O | $5_b$ | $GP_{dil+stat}$ | $0.602 \pm 1.83 \times 10^{-2}$ | $5_p$ | $<1 \times 10^{-4}$ * | $0.446 \pm 7.70 \times 10^{-3}$ | $5_p$ | $<1 \times 10^{-4}$ * |
| I | $5_p$ | $GP_{dil+stat}$ | $0.947 \pm 3.77 \times 10^{-4}$ | 3 | $<1 \times 10^{-4}$ * | $0.627 \pm 4.96 \times 10^{-4}$ | - | - |
| I | $5_b$ | $GP_{dil+stat}$ | $0.729 \pm 5.70 \times 10^{-3}$ | $5_p$ | $<1 \times 10^{-4}$ * | $0.544 \pm 6.40 \times 10^{-3}$ | $5_p$ | $<1 \times 10^{-4}$ * |

* null hypothesis (model performances were not different) rejected at the $p < 0.05$ level.

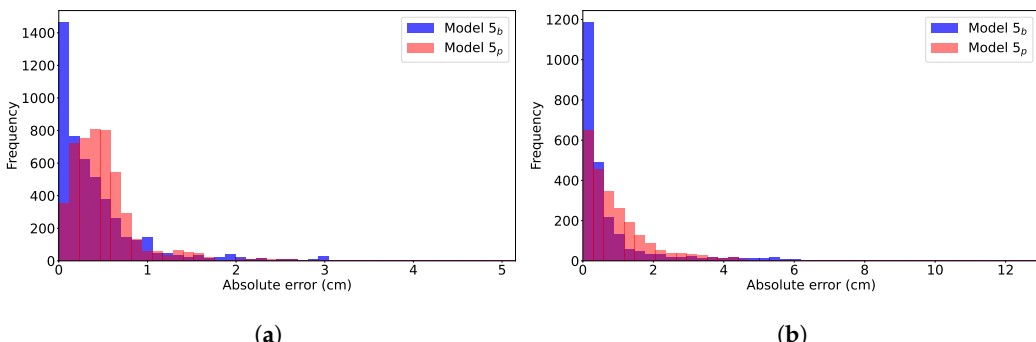

**Figure 4.** Comparison of the distribution of $GP_{dil+stat}$ prediction errors for models $5_p$ and $5_b$ for (**a**) one fold of the Ohio data and (**b**) the independent test dataset.

In the comparative analysis presented in Figure 5, we observed distinct differences in the values of the RMSE values of the cross-validation test dilation RMSE values between models $GP(t)_{dil}$ and $GP_{dil+stat}$ as a function of the elapsed hours since the previous exam. The errors were the lowest for the predictions that considered less than 3 h of elapsed time, which represents approximately 90% of the test data. $GP(t)_{dil}$ exhibits an RMSE range from 2.3 to 4.5, with gradually increasing error as the elapsed hours increase. In contrast, $GP_{dil+stat}$ presents a more compressed RMSE range, fluctuating between 0.9 and 1.65. This model also displays a consistent upward trend in RMSE mean and range as hours increase, but the progression is subtler than $GP(t)_{dil}$. The tighter RMSE range in $GP_{dil+stat}$ at each hour indicates that its better predictive accuracy is more stable across folds.

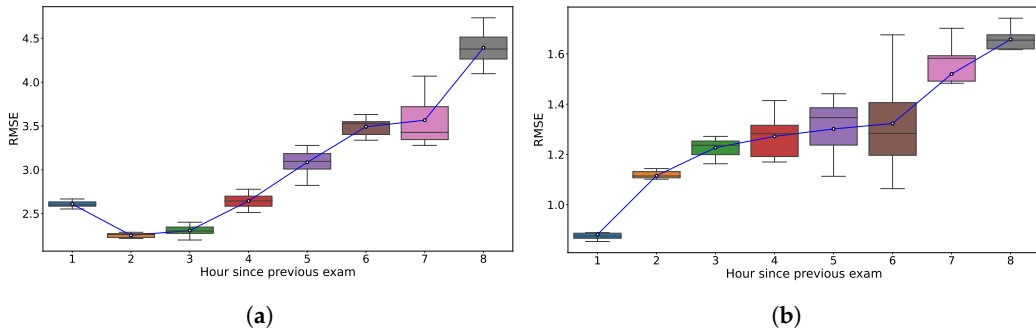

**Figure 5.** Visualization of box plot of dilation RMSE as a function of time since previous exam categories for the (**a**) $GP(t)_{dil}$ and (**b**) $GP_{dil+stat}$ models. Each box represents the RMSE interquartile range, with the white dot indicating the mean RMSE for each category, aggregated over 10 folds on the Ohio data.

In the comparison shown in Figure 6, we observe the predictive capacities of different models in estimating cervical dilation during labor. In particular, Model $5_p$, as shown in Figure 6b, demonstrates consistency in its predictions across deliveries in the test dataset. Its predictions show a tightly clustered range of standard deviations, narrowly ranging from approximately 0.652 to 0.656.

In Figure 6a, we can see $ME_{dil}$, which has a narrow range of standard deviations from 1.116 to 1.128. On the other hand, Figure 6c shows Model $5_b$, which exhibits more variability than $ME_{dil}$. Model $5_b$ shows a gradual increase in standard deviation over time, moving from about 0.48 to 0.58 as the time from the previous exam extends from 1 to 8 h.

Both models $5_p$ and $5_b$ show trends of increasing mean standard deviation as the time interval between exams increases. However, it should be noted that model $5_b$ demonstrates a more rapid increase in standard deviation compared to Model $5_p$. This suggests that unpredictability escalates more quickly over time in Model $5_b$ than in Model $5_p$.

### 3.6. Computational Load

Finally, we measured the computational load of training and inference with the GP models $5_b$ and $5_p$. Training times to generate the $5_p$ and $5_b$ models in each fold were $14.0 \pm 4.5$ h (mean $\pm$ std) on a machine with an NVidia Tesla T4 GPU with 16 GB of memory. Inference times were estimated on a standard laptop with CPU computation, using 100 deliveries (370 exams total) from one fold of cross-validation for the Ohio dataset. The average inference times per exam for $5_p$ and $5_b$ were 114 ms and 115 ms, respectively. The corresponding times per delivery were 423 ms and 427 ms, respectively.

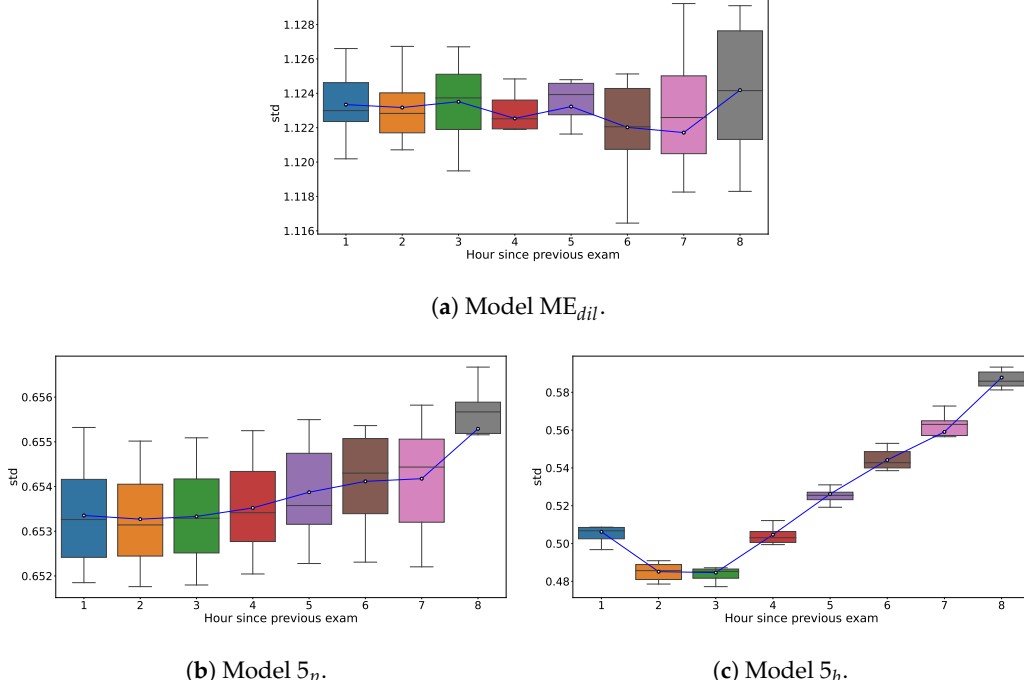

(**a**) Model $\mathrm{ME}_{dil}$.

(**b**) Model $5_p$.

(**c**) Model $5_b$.

**Figure 6.** Box-plot representation illustrating the variation in the standard deviation of dilation predictions concerning elapsed time since the last pelvic examination, for models (**a**) $\mathrm{ME}_{dil}$, (**b**) $5_p$, and (**c**) $5_b$. Each box represents the interquartile range of the credible interval, the white dot indicating the mean credible interval for each category, aggregated over 10 folds.

## 4. Discussion

As discussed in the Section 1, the two main models of labor progress in clinical use are based on the univariate time models of Friedman and Kroll [4] and Zhang et al. [5]. In this study, we used a univariate GP model $\mathrm{GP}(t)_{dil}$ as a baseline time-based model. In fact, given the restricted information content of a time-based univariate regression framework, in experimentation with our data, we found that all the models we tested had similar prediction errors, whether they were GP, or an eighth-order polynomial model similar to that used by Zhang et al. [5]. However, it should be reiterated that the Zhang et al. [5] model was estimated using relative time to delivery, while we used $t_0$-based time to generate causal models; see Hamilton et al. [21] for this detailed comparison. This finding underscores the limitations inherent in univariate models when applied to the multifaceted nature of labor progression.

However, with the selection of clinically relevant predictors in a multivariate model that we have specified in this study, each of which is available in data from the EMR, and from labor data in routine pelvic examinations and the EFM, the prediction error was significantly reduced compared to the univariate approach when using the $\mathrm{ME}_{dil}$ model. Furthermore, while the single-prediction model $\mathrm{GP}_{dil}$ performed equivalently to $\mathrm{ME}_{dil}$, it was superior, with statistical significance, for the independent test set, i.e., for data from a different geographical and clinical setting, and not used during the training of the model.

The multivariate model $\text{GP}_{dil+stat}$ had a prediction horizon that was remarkably steady even for up to 6 h intervals between exams (see Figure 5). Importantly, however, the predictor set did not include time, as it did not provide a significant improvement on the RMSE error performance metric. One desirable aspect of GP modeling is that it inherently provides estimates of predictor importance on a per-kernel basis. In Hamilton et al. [21] we provide full quantitative details; we summarize these results here to better interpret this study. The cumulative contraction count was the most predictive regressor in conjunction with the Gaussian kernel, which models the smooth continuity of the prediction. Thus, we may consider contraction count as a surrogate for time that better captures the degree of contraction-related stimuli that impinge on the cervix. Therefore, under normal physiological conditions and within reasonable limits, more contraction should promote labor progression. On the other hand, the previous dilation was the most predictive regressor for the Matérn kernel, which allows for abrupt prediction changes. This is consistent with a distinctive non-linear profile that is commonly observed in the cervical dilation progression, especially towards the end of labor.

The superiority of the dual prediction of dilation and station in the $\text{GP}_{dil+stat}$ model compared to the single prediction models $\text{GP}_{dil}$ and $\text{GP}_{stat}$ showed that our use of the coregionalization kernel successfully incorporated output correlations to better predict each output. This is especially relevant to our problem, where cervical dilation and fetal station are such interacting factors. The ability to improve the respective predictions is a confirmation of the results in Bonilla et al. [26], which considered tasks outside the biomedical domain, and speaks to the broad applicability of this approach to other similar biomedical problems that require multiple related forecasts.

The final step in the modeling of test set adaptation, shown in the black lines and surrounding blue bands of our figures, improved prediction accuracy for all of our dataset outputs. But equally importantly, they provided confidence bounds on our estimates that were tailored to the particular delivery trajectory. This is in contrast to the fixed confidence bounds of a parametric model like ME. The variation of the standard deviation of models $5_p$ and $5_b$ with increasing time between exams show that the model uncertainty adapts to the data. This trend is characteristic of most time series models, reflecting an increase in uncertainty with longer gaps between observations. It is also clinically reasonable that such uncertainty about predicted dilation increases with these longer gaps. In the real-time setting, where the potential for alarm fatigue among clinical staff is great, this gives the clinician better grounds for heeding or ignoring the model prediction.

Finally, we reiterate that we frame this problem as single-step prediction of the current exam using input data from the immediate past. However, with small adjustments to the model, it is also possible to perform multi-step predictions into the future. But such forecasting is fraught with other uncertainties that may be unrelated to the inherent model uncertainty; it may also open possibilities for misuse and unnecessary interventions. Furthermore, our prediction model matches clinical behavior, where the expected progress based on previous information is compared to the latest data available at each exam, in order to make decisions concerning intervention.

A limitation of this study, and all other studies that produce a labor curve, is related to initial data selection. All labor curve studies are based on data from women with a vaginal birth, that is, women who did not have a cesarean delivery. Cesarean rates can vary several fold from region to region such that the residual vaginal delivery group may represent a very different proportion of parturients. Medical and anthropometric characteristics can also be very different across regions affecting the applicability of these models. In addition, it must not be assumed that a certain deviation from expectation, for example below the $5^{th}$ percentile is necessarily abnormal. By definition, 5% of examinations from normal labors exhibited such results at times. Determining intervention thresholds requires careful study of both normal and clearly abnormal labors to define optimal criteria.

## 5. Conclusions

We have presented a novel Gaussian process model of labor progress, and have demonstrated this it is suitable for prediction inference in real-time, which is an important requirement for clinical utility. The model predicts cervical dilation and fetal station based on clinically relevant predictors available from the electronic medical record, the pelvic exam, and cardiotocography. We show that the model is more accurate than a commonly used statistical approach, but additionally, it provides confidence estimates on the prediction that are calibrated to the specific delivery. Finally, we show that predicting both dilation and station with a single Gaussian process model was often more accurate than two separate models with single predictions.

**Author Contributions:** Conceptualization, E.F.H. and P.A.W.; methodology, T.Z., E.F.H. and P.A.W.; software, T.Z. and P.A.W.; validation, T.Z., E.F.H. and P.A.W.; formal analysis, T.Z., E.F.H. and P.A.W.; investigation, T.Z., E.F.H. and P.A.W.; resources, E.F.H. and P.A.W.; data curation, E.F.H. and P.A.W.; writing–original draft preparation, T.Z., E.F.H. and P.A.W.; writing–review and editing, T.Z., E.F.H. and P.A.W.; visualization, T.Z.; supervision, E.F.H. and P.A.W.; project administration, E.F.H. and P.A.W.; funding acquisition, E.F.H. All authors have read and agreed to the published version of the manuscript.

**Funding:** PeriGen, Inc., a Halma company, provided the financial support for the infrastructure required to conduct this research.

**Institutional Review Board Statement:** This study was considered exempt by the Institutional Review Boards at OhioHealth (State of Ohio), Wayne State University (Detroit, MI), and Legacy Health (States of Washington and Oregon).

**Informed Consent Statement:** All data was de-identified. The authors obtained permissions to use the data for this project from the hospital systems that provided the data.

**Data Availability Statement:** The data from this study is not available.

**Acknowledgments:** The authors would like to thank Kevin Flores for his constructive feedback during the preparation of this manuscript.

**Conflicts of Interest:** The authors report no conflict of interest. E.F.H., P.A.W., T.Z. report being employed by PeriGen, Inc.

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
