# Peer review of "Data-Driven Insights into Labor Progression with Gaussian Processes"

_bioengineering, doi:10.3390/bioengineering11010073_

Round 1

Reviewer 1 Report

Comments and Suggestions for Authors

Report on "Data-Driven Insights into Labor Progression with Gaussian Processes" by Zhoroev et al.

The manuscript introduces a novel Gaussian process model for labor progression, utilizing a Bayesian framework that aligns well with the inherent characteristics of the data. The methodology employed in the modeling process is meticulously detailed, presenting a well-structured manuscript. However, certain minor concerns warrant consideration.

1. Computational Complexity of Gaussian Process Models:

The authors acknowledge a critical issue associated with Gaussian process (GP) models, namely their cubic scaling computational complexity in relation to the dataset size. It would be beneficial for the reader if the authors could provide information on the time required for training the models. Addressing this concern is pertinent, as the practicality of applying complex Bayesian models hinges significantly on computational efficiency. A disclosure of the training duration would enhance the comprehensibility of the article and aid practitioners in assessing the feasibility of implementing similar models.

2. Specification of Random Effects in Mixed Models:

A second concern involves the need for a more explicit specification of the random effects utilized in the mixed models. Clarity regarding the random effects is crucial for facilitating the replication of similar modeling approaches by other researchers. The inclusion of such details enhances the transparency and reproducibility of the study, enabling the scientific community to build upon the proposed methodology effectively.

Author Response

Please see attached pdf file with response to all reviewers.

Reviewer 2 Report

Comments and Suggestions for Authors

The paper is about using Sparse Gaussian Process Regression to predict labor progression (cervical dilation and fetal station) based on delivery data obtained from cardiotocography. The introduction part is precise; analysis and the conclusions drawn from the results are thorough. The described methodology appears well-considered and thorough, addressing key aspects from model architecture and training to testing and evaluation. The integration of advanced optimization algorithms and the hybrid optimization strategy adds a layer of sophistication to the approach. The use of a meta-learning framework further enhances the model's adaptability to new data scenarios. The labels in the figures are slightly unclear especially figure 2. Discussion part of the paper could also talk about the limitations of the methodology, and further explore the applicability of SPGR in the field of biomedical research. Conclusion is concise, but it does not conclude the paper that well, incorporating all the sections of the paper in the conclusion can make it more comprehensive.  

Comments:

1.       Page 5 Line 219: The topic does not fit the content; the paragraph discusses input variables used for the model implementation. The justification of which is lacking in the paper.

2.       Page 6 Line 262 Normal labor and labor arrest should be explained in further detail.

3.       Figure 2, The figure labels are misleading, which ones are labor arrest data and normal level data? Also, does the graph show prediction for only dilation, but the label says it’s predicting fetal station as well

4.       Page 6 Line 258 asserts that the predictors were selected from the population level analysis in the next section (3.2), whereas no justification is provided for the selection in the section.

5.       Page 8 Table 1 Mixed Effects model hasn’t been used for the fetal station data; the reason is not mentioned.

6.       Page 8 Line 294 “ In particular, while station prediction for GPdil+stat compared to GPdil” . Comparison to GPstat was not statistically significant. (Typographical error)

7.       Page 8 Line 302 “… immense potential as a tool for real- world applications” The sentence belongs to the discussion section as it is inferring from the results; also, the real-world applications should be discussed further in order to provide more context for this assessment

Comments on the Quality of English Language

Only minor issues, can be caught with a quick review.

Author Response

(The authors gave the same response as above.)

Reviewer 3 Report

Comments and Suggestions for Authors

The paper introduces a novel Gaussian process (GP) model aimed at improving the prediction of labor progression, specifically cervical dilation and fetal station, using data typically gathered from pelvic exams and cardiotocography. This model is presented as an alternative to the commonly used time-based models for assessing labor progression, which have not been effective in reducing cesarean-section rates. This paper is well-written with a clear structure. However, I have some concerns that I believe should be resolved before publication.

  • The paper should clarify whether the predictive modeling approach involves multi-step predictions or is limited to one-step-ahead predictions. This distinction is important for understanding the model's capabilities and practical applications in real-time clinical decision-making. Multi-step prediction models can provide a more comprehensive outlook for labor progression, which could be critical in a clinical context.
  • A clearly articulated problem formulation is essential. The paper should explicitly state the inputs and outputs of the proposed system. For instance, the inputs could be clearly defined as clinical parameters obtained from pelvic exams and cardiotocography, while the outputs might be the predicted cervical dilation and fetal station. This clarity will ensure that readers can immediately grasp the primary focus and contributions of your research.
  • The description of the Mixed-effects (ME) model utilized in the study is currently lacking in depth. It is crucial to elaborate on how ME models are integrated with your GP approach. This is not just about which software package was used, but rather how the ME model contributes to the overall modeling framework, its role in the methodology, and its impact on the results. Detailing this integration would strengthen the methodological rigor of the paper and provide a clearer understanding of the approach taken.
  • The current literature review on Gaussian Processes (GP) appears to be broad and somewhat divergent from the core subject of the paper. It would be beneficial to focus the review more tightly on applications of GP specifically within the biological context, particularly in obstetrics or related medical fields. This shift in focus could help to underline the relevance and novelty of your approach, emphasizing how GP can address specific challenges in biological data analysis.

Author Response

(The authors gave the same response as above.)

Round 2

Reviewer 1 Report

Comments and Suggestions for Authors

All my concerns have been addressed by the authors.

Reviewer 3 Report

Comments and Suggestions for Authors

the author has resolved my major concerns. Nonetheless, I feel that the one-ahead prediction has limited applications in reality.